# Implementation of a Cryptographic Hash Function Based on a Deep Neural Network

Dmitry V. Iatsenko
Southern Federal University
Rostov-on-Don, Russia
d.yacenko@gmail.com

Denis S. Gorin
RTU MIREA
Moscow, Russia
gorin@mirea.ru

## Abstract

We present a two-layer construction for image hashing. First, a perceptual binary code $c(x)$ is derived from a ResNet-18 embedding (after global average pooling, $d = 512$) via a linear projection and sign quantization; optionally, a real-valued serialization of length $n = 8ds$ bits is used. The code $c(x)$ enables fast approximate nearest-neighbor search. Second, $c(x)$ serves as a noisy source for a fuzzy extractor producing a reproducible secret $R$ and public data $P$; a cryptographic tag $T$ is then derived via KDF and HMAC/SHA-3. This preserves similarity search over $c(x)$ while assigning cryptographic guarantees (preimage/second-preimage/collision) to $T$, which reduce to the security of the underlying primitives given sufficient post-publication min-entropy $H_\infty(C \mid P)$. We also report preliminary proof-of-concept experiments on a small demo dataset: we measure intra-BER under admissible transforms, inter-image Hamming distances, and successful FE/tag verification on retrieved candidates. These experiments are intended to validate the mechanics of the proposed pipeline rather than provide a large-scale benchmark.

Keywords: perceptual hash; ResNet-18; feature binarization; fuzzy extractor; min-entropy; approximate nearest neighbor; HMAC; SHA-3; error-correcting codes; cryptographic tag.

## 1 Introduction

Image matching systems often require two different properties at once. On the one hand, they must support similarity search: visually similar images should map to nearby codes. On the other hand, they may require a stronger verification mechanism that is stable under admissible distortions but still grounded in standard cryptographic primitives. These two goals are different in nature: perceptual hashing is useful for retrieval, while cryptographic hashing is useful for strict verification.

This paper studies a two-layer construction that separates these roles. First, a deep visual embedding is converted into a binary perceptual code suitable for approximate nearest-neighbor search in Hamming space. Second, this code is treated as a noisy source for a fuzzy extractor, from which we derive a reproducible secret and a cryptographic tag. Thus, the perceptual layer supports search, while the cryptographic layer supports verification.

The main contribution of the paper is a compact implementation-oriented formulation of this pipeline together with a preliminary proof-of-concept evaluation. We explicitly distinguish what is claimed at the perceptual-code level and what is claimed only after the fuzzy-extractor and tag-generation stages. In particular, we do not claim that the perceptual code itself is a cryptographic hash; rather, cryptographic guarantees are assigned only to the final tag.

Terminology and metrics. Throughout the paper, BER denotes the bit error rate between two binary codes. FRR (false reject rate) is the probability that a genuinely matching query fails verification. FAR (false accept rate) is the probability that a non-matching query is

accepted. We also briefly refer to the classical cryptographic criteria SAC (strict avalanche criterion) and BIC (bit independence criterion) only for contrast with perceptual hashing; they are not required of the perceptual code used in our construction.

## 2 Perceptual Hash Layer

### 2.1 Perceptual hash based on ResNet-18

We consider an embedding based on the ResNet-18 convolutional network with the final fully connected layer removed (He et al., 2016). Let $e(x) \in \mathbb{R}^d$ be the output feature vector after global average pooling (for the standard ResNet-18, $d = 512$ is typical). We then construct from $e(x)$ a perceptual hash intended for searching visually similar images.

**Canonical preprocessing.** For reproducibility we fix a pipeline: resizing to $256 \times 256$, center crop $224 \times 224$, channel normalization $(\mu, \sigma)$, and fixed byte order and image format. This minimizes feature drift due to input ambiguities.

**Definitions.** We use two representations:

1. Real-valued descriptor $v(x) = \mathrm{norm}(e(x)) \in \mathbb{R}^d$ ($L_2$ normalization; optionally whitening/PCA).
2. Binary perceptual hash $c(x) \in \{0,1\}^n$—a quantized code for fast comparison in Hamming space.

**Binary hash construction.** Let $A \in \mathbb{R}^{n \times d}$ and $b \in \mathbb{R}^n$ be fixed public projection/centering parameters chosen on a training set. Possible options include random hyperplanes (p-stable LSH (Datar et al., 2004)), learned orthogonal/projection schemes (ITQ (Gong & Lazebnik, 2011)), and quantization from the PQ/OPQ family for compact indices (Jégou et al., 2011; Ge et al., 2013). Define $z(x) = A\,v(x) + b$ and sign binarization

$$c_i(x) = \mathbf{1}\{z_i(x) \geq 0\}, \quad i = 1, \ldots, n.$$

Typically $n \in [256, 2048]$: increasing $n$ improves discriminativeness, but increases the average BER between "related" instances, which is important for the subsequent use of a fuzzy extractor.

**Remark on serialization.** If needed, the real-valued descriptor $v(x)$ is stored in a fixed format (float16/float32); its length is $8ds$ bits. This is not a cryptographic hash, but an auxiliary representation for search/re-quantization.

### 2.2 Properties of the perceptual hash

**Stability (robustness).** A family of admissible transforms $\mathcal{T}$ is specified (moderate geometric/photometric distortions, compression). For $T \in \mathcal{T}$ we evaluate $\mathrm{BER}(x, Tx)$. We require $\mathbb{E}_{x,T}[\mathrm{BER}(x, Tx)] \leq \varepsilon$ (e.g., $< 0.1$).

**Discriminativeness.** For pairs of "different" images $(x, x')$, the distribution of $\mathrm{Ham}(c(x), c(x'))$ should be substantially shifted to the right compared to "same/similar" pairs; we evaluate ROC/mAP for ANN search in Hamming space (Gong & Lazebnik, 2011; Jégou et al., 2011; Ge et al., 2013).

**Bit balance and weak inter-bit correlations.** We would like $\Pr[c_i = 1] \approx \frac{1}{2}$ and small $|\mathrm{corr}(c_i, c_j)|$ for $i \neq j$. Balance can be achieved by selecting offsets $b$ and/or thresholds on data.

**Collisions (perceptual).** Collisions are expected and acceptable: code matches should, with high probability, correspond to perceptually similar images. The theoretical estimate $\approx 2^{-n}$ applies only for independent balanced bits; in practice we use empirical frequencies.

Hardware-software stability. To reproduce the binary code we fix the model/library versions and numeric precision; sign binarization after a linear projection reduces sensitivity to floating-point implementation details.

## 2.3 Limitations and threat model

The perceptual hash $c(x)$ is not a cryptographic hash in the sense of (Dang, 2012; Menezes et al., 1996): it is deterministic, unkeyed, allows targeted collisions, and is vulnerable to adversarial modifications that preserve the code (Struppek et al., 2021). Pre-image/second-pre-image/collision resistance properties are not claimed at this level.

## 2.4 Preparing to apply a fuzzy extractor

In the second part, we build a fuzzy extractor on top of $c(x)$ (Dodis et al., 2008). Already at this stage we fix the source characteristics required for the FE:

- Bit error for "similar" instances: we estimate the distribution of $\mathrm{BER}(x, Tx)$ and choose $n$/thresholds so that an error-correcting code capable of correcting $t$ errors exists with acceptable false reject rate (FRR) and false accept rate (FAR).
- Min-entropy: we empirically estimate a lower bound on $H_\infty(C)$ and approximate independence/balance of bits to ensure sufficient entropy of the extracted secret.
- Public data: parameters $(A, b)$ and FE sketch data are public; we then analyze leakage $H_\infty(C \,|\, P)$.

## 2.5 Selective sensitivity: robustness within class and "avalanche" between classes

The classical cryptographic criteria—the strict avalanche criterion (SAC) and the bit independence criterion (BIC)—(Webster & Tavares, 1986; Cusick, 1994; Menezes et al., 1996) require that flipping a single input bit changes, on average, about half of the output bits. This objective is relevant for cryptographic hashes, but it does not coincide with the goals of perceptual hashing, where the priority is stability to small perceptual transforms and separability of dissimilar images. Therefore, below we use the term "selective sensitivity" instead of "avalanche effect" and distinguish two regimes:

(i) Robustness within class. We specify a family of admissible transforms $\mathcal{T}$ (moderate geometric/photometric changes, compression, etc.). For $T \in \mathcal{T}$ we evaluate

$$\mathrm{BER}_{\mathrm{intra}}(x, Tx) \;=\; \frac{1}{n} \sum_{i=1}^{n} \mathbf{1}\{c_i(x) \neq c_i(Tx)\}.$$

We require $\mathbb{E}_{x,T}[\mathrm{BER}_{\mathrm{intra}}] \leq \varepsilon$ for small $\varepsilon$ (e.g., $< 0.1$), ensuring code reproducibility for similar images. This quantity is critical for the subsequent fuzzy extractor, since it determines the required error-correction capability (radius $t$).

(ii) "Avalanche" between classes. For pairs of dissimilar images $(x, x')$, the classical SAC/BIC ideal for cryptographic hashes would correspond to an almost random behavior with normalized Hamming distance near $\frac{1}{2}$. Our perceptual code does not aim to satisfy this strict cryptographic criterion literally; rather, we seek an avalanche-like between-class effect: dissimilar images should induce substantially larger Hamming distances than near-duplicate or admissibly transformed pairs, thereby reducing the probability of false matches.

Practical check. We run two experiments:

- Intra robustness curve: for each test $x$ and a set of $T \in \mathcal{T}$, we build a histogram of $\mathrm{BER}_{\mathrm{intra}}$; we report $\mu, \sigma$ and the quantile $q_{0.95}$—this directly sets the required correction radius $t \approx n \cdot q_{0.95}$ for the FE.
- Inter separability curve: for random pairs $(x, x')$, we build the distribution of $\mathrm{Ham}(c(x), c(x'))/n$; ideally, the median is close to 0.5 and the mass near $[0, 0.3]$ is small.

Bit balance and weak correlation. To increase the entropy of the source $C = c(X)$, we prefer $\Pr[c_i = 1] \approx \frac{1}{2}$ and small $|\mathrm{corr}(c_i, c_j)|$ for $i \neq j$; this is achieved by selecting offsets/thresholds and (if needed) orthogonal projections (Datar et al., 2004; Gong & Lazebnik, 2011). These properties matter not only for search quality, but also for a lower bound on the min-entropy used by the fuzzy extractor (Dodis et al., 2008).

Remark on vulnerabilities. Neural-network codes are vulnerable to targeted adversarial changes that can preserve or imitate the code (Struppek et al., 2021); this is an expected limitation of perceptual hashing and does not contradict the stated goals. Cryptographic security is provided at the next step by the fuzzy extractor and the subsequent use of standard cryptographic primitives (KDF and AEAD). Therefore, at the perceptual-hash level it is sufficient to require robustness to uncontrolled input distortions, rather than cryptographic resistance.

## 2.6 Perceptual collisions and empirical separability

Within perceptual hashing, by a collision we mean equality of binary codes $c(x) = c(x')$ or their "sticking" within a Hamming radius $r$, $\mathrm{Ham}(c(x), c(x')) \leq r$, for different and dissimilar images $x \neq x'$. The goal is for such events to be rare, and for the distance distribution for dissimilar pairs to be shifted to the right relative to near-duplicate pairs, with the idealized random-hash regime near $n/2$ serving only as a conceptual reference point.

Metrics.

1. Inter-distribution: the distribution $D_{\mathrm{inter}}$ of $\mathrm{Ham}(c(x), c(x'))/n$ for random dissimilar pairs $(x, x')$. Ideally, the median $\approx 0.5$ and the left mass is small.

2. Probability of a random match under threshold $r$: $p_{\leq r} = \Pr[\mathrm{Ham}(c(x), c(x')) \leq r]$ under $D_{\mathrm{inter}}$. For a collection of size $M$, the expected number of false matches is $\mathbb{E}[\mathrm{FP}] \approx \binom{M}{2} p_{\leq r}$.

3. Balance and correlations: we empirically estimate $\Pr[c_i = 1]$ and $\mathrm{corr}(c_i, c_j)$. Balance $\approx 1/2$ and weak inter-bit correlations increase the source entropy and reduce the risk of perceptual collisions (Gong & Lazebnik, 2011; Jégou et al., 2011; Ge et al., 2013; Datar et al., 2004).

Why we do not use the "birthday bound" directly. The classical bound $2^{n/2}$ for collision search holds for an ideal random hash into $\{0, 1\}^n$ (Menezes et al., 1996). Our code $c(x)$ is constructed from an embedding and a thresholded linear projection; bits can be biased and dependent. Therefore, we rely on empirical $D_{\mathrm{inter}}$ and $p_{\leq r}$ rather than an IID model.

Attacks and assumptions. In a white-box setting, targeted construction of collisions for $c(\cdot)$ via optimization (adversarial techniques) is possible, consistent with known vulnerabilities of neural perceptual features (Struppek et al., 2021). This does not contradict the purpose of this section: here we evaluate perceptual properties and the probability of random matches. Cryptographic collision resistance is provided at the next layer (see below).

Relation to the fuzzy extractor. To build an FE we need: (i) low intra-BER for "own" pairs (see the previous section) and (ii) sufficient min-entropy of the source $C = c(X)$. Min-entropy is estimated from code/cluster frequencies and bit balance; then the extracted secret length $R$ is constrained by standard relations (residual randomness after publishing helper data), see (Dodis et al., 2008). Collisions $c(x) = c(x')$ for dissimilar $x, x'$ are acceptable to the extent that they are rare and do not reduce the min-entropy below the threshold required for the target key length.

## 3 Cryptographic tag over a perceptual hash via a fuzzy extractor

In the previous sections we constructed a binary perceptual code $c(x) \in \{0, 1\}^n$ with low intra-BER for "own" transformations and an avalanche-like right-shifted inter-distance dis-

tribution for "others". We now use it as a "noisy source" in a fuzzy extractor (FE) (Dodis et al., 2008) to obtain a reproducible secret $R$ and public helper data $P$, and then produce a cryptographic tag $T$.

## 3.1 Construction scheme

Components.

- Perceptual code: $C = c(X) \in \{0,1\}^n$ (see above).
- FE algorithms: Gen : $\{0,1\}^n \to \{0,1\}^\ell \times \mathcal{P}$ and Rep : $\{0,1\}^n \times \mathcal{P} \to \{0,1\}^\ell$ such that, if $\text{dist}_H(C, C') \leq t$, then with high probability $\text{Rep}(C', P) = R$ where $(R, P) = \text{Gen}(C)$ (Dodis et al., 2008).
- Extractor/KDF: KDF (e.g., HKDF over SHA-3).
- Tag primitive: HMAC or SHA-3 (depending on the application) (Menezes et al., 1996).

Enrollment.

1. Compute $c = c(x)$.
2. $(R, P) \leftarrow \text{Gen}(c)$.
3. Compute the key $K \leftarrow \text{KDF}(R \,\|\, \text{ctx})$ with domain separation string ctx.
4. Form the record tag $T \leftarrow \text{SHA3-256}(K \,\|\, \text{meta})$ or $T \leftarrow \text{HMAC}_K(\text{meta})$.

Store in the database: the index code for search $c$ or its compact/LSH representation, the public data $P$, and the tag $T$. (Under stricter privacy requirements one may store only LSH signatures instead of the full $c$.)

Query (search/verification).

1. For a query $x'$, compute $c' = c(x')$.
2. Similarity search: use the index (Hamming/ANN) to find candidates with $\text{dist}_H(c', c) \leq r$.
3. Candidate verification: for each returned record, reconstruct $R' = \text{Rep}(c', P)$, then compute $K' = \text{KDF}(R' \,\|\, \text{ctx})$ and $T'$, and compare $T'$ with the stored $T$; equality confirms "identity by tag".

## 3.2 Properties and guarantees

Compatibility with search. Indexing and nearest-neighbor selection are performed using the perceptual code $c$ (or LSH), i.e., the ability to find similar images is preserved. The FE/cryptographic part is used only at the candidate verification stage and does not "break" the metric.

Cryptographic security of the tag. The tag $T$ is built from $K = \text{KDF}(R, \text{ctx})$ where $R$ is the FE output. Under sufficient source min-entropy after publishing $P$ (denote $\tilde{H}_\infty = H_\infty(C \,|\, P)$), the length $\ell$ is chosen with a margin according to standard FE estimates (see (Dodis et al., 2008)). Then the security of $T$ (preimage/second-preimage/collision) reduces to the security of the underlying primitive (SHA-3/HMAC) and the unpredictability of $R$:

- Collision resistance: finding two inputs that produce the same $T$ is no easier than breaking the collision resistance of the chosen primitive (for fixed ctx/meta).
- Preimage/2nd-preimage: recovering $R$ from $T$ or finding another $R'$ with the same $T$ reduces to breaking the primitive; obtaining a corresponding $c'$ additionally requires meeting the FE tolerance (otherwise Rep will not reconstruct the original $R$).

Important: "tag equality" does not mean bitwise equality of the original pixels; the FE intentionally allows variations controlled by the threshold $t$.

Parameters and code selection.

- We choose the length $n$ and binarization thresholds from measured BER distributions so that the 95th percentile of intra-BER fits into the admissible radius $t/n$ for the selected error-correcting code (BCH/LDPC/RS).

- We choose the secret length $\ell \leq \tilde{H}_\infty - \mathrm{losses} - \mathrm{safety}$ (a margin for leakage through $P$ and statistical deviations), as recommended by FE theory (Dodis et al., 2008).

- Domain strings ctx and meta (e.g., application/version/policy identifiers) prevent cross-domain linkage and enable rotation.

Confidentiality and linkability. Published $P$ does not reveal $R$ by definition (Dodis et al., 2008), but it can facilitate record linkability. To reduce risks: (i) include a random seed/version in $P$ (different instances yield different $P$); (ii) apply domain separation in KDF; (iii) where possible, store LSH signatures instead of the full $c$ in the index.

Fault tolerance. Observed tails of intra-BER are compensated by a margin in $t$ and by the code choice; under degraded conditions (strong distortions) a record may fail to reconstruct (FRR). This is controlled by selecting $n, t$ and by the profile of admissible transforms $\mathcal{T}$.

### 3.3 Overall architecture

$$
\begin{array}{lll}
\text{Image } x & \xrightarrow{\ e(x)\ } & \text{embedding} \to \text{binarization} \to c(x) \\
& & \quad \downarrow \text{ Gen (FE)} \Rightarrow (R, P) \\
& & \quad \downarrow \text{ KDF} + \text{HMAC/SHA-3} \Rightarrow T \\
\text{Stored in DB} & : & \text{index: } c \text{ (or LSH), public: } P, \text{ cryptographic tag: } T \\
\text{Query } x' & \xrightarrow{\ c(x')\ } & \text{Hamming search} \Rightarrow \text{candidates} \Rightarrow \mathrm{Rep}(c', P) \Rightarrow R' \Rightarrow T' \text{ compare with } T
\end{array}
$$

A reference implementation is available in a public repository: https://github.com/d-yacenko/Article14.git.

### 3.4 Limitations and practical notes

- The perceptual part remains vulnerable to targeted (adversarial) manipulations; tag security depends on FE+primitive, not on an "avalanche" property of the embedding (Struppek et al., 2021).

- For exact byte-level identity one may additionally store a classical hash (e.g., SHA3-256 of a canonicalized file); it is not used for search, but provides strict equality checks.

- Debugging/compatibility: fix model/library versions; use domain strings and version parameters $(A, b)$ and the FE instance.

Thus, the proposed construction preserves similarity search at the perceptual-code level and adds on top a cryptographic verification tag with guarantees grounded in the fuzzy extractor and standard cryptographic primitives (Dodis et al., 2008; Menezes et al., 1996).

### 3.5 Use in two roles: search and steganography triage

The pipeline perceptual hash $\to$ fuzzy extractor (FE) addresses the main task of searching and verifying similar images under small distortions: the perceptual hash preserves similarity-aware retrieval, while the FE (Dodis et al., 2008) stabilizes a "noisy" bit fingerprint into a deterministic key/tag (Dang, 2012; Menezes et al., 1996).

Additional role: an indicator for steganography triage. We note an auxiliary signal that can be useful for investigations and security triage. If, for a pair of images, we simultaneously observe:

1. high perceptual similarity (small pHash distance) and successful reproduction in the FE (match=True), and

2. the files differ under byte-level comparison (including differences at the container/metadata level),

then such a pair can be assigned to the class "suspicious, requiring additional checks for steganography". The intuition is simple: steganographic methods often introduce small, visually imperceptible modifications (LSB/DCT/chunks) that do not destroy pHash and remain within the FE tolerance, but still change the byte representation. In the reference implementation https://github.com/d-yacenko/Article14.git, we provide an example: the original file images/0.07180100_1442390923_.jpg and the file images/0.07180100_1442390923_stegano.jpg that contains steganographic data (the string hello world!) are identified as similar by the proposed mechanism, yet differ even in size. This demonstrates the indicator.

**Important limitation.** This signal is a weak indicator: benign operations behave the same way (JPEG re-encoding, resize, crop, EXIF/ICC editing, auto-rotation, recompression, etc.). Therefore, this criterion is neither necessary nor sufficient to conclude the presence of steganography; it should be treated as a trigger for additional diagnostics.

**Recommended triage pipeline.** After the indicator triggers, perform lightweight automated checks:

1. Normalization and re-evaluation: strip metadata, convert images to a common format/size/quality, then recompute pHash and the FE; persistence of the match increases priority.

2. Container analysis: for JPEG, inventory APP/COM segments and tail after EOI; for PNG, check nonstandard ancillary chunks (iTXt/zTXt/custom) and anomalies in IDAT size.

3. Quick statistical tests: simple LSB/$\chi^2$/RS tests for uncompressed/PNG, and a basic analysis of DCT coefficient distributions for JPEG (as probabilistic screening, not as proof).

This block does not change the cryptographic part; it only formalizes a "suspicion signal" that naturally arises from properties of pHash and the FE (Dodis et al., 2008; Dang, 2012; Menezes et al., 1996) and can be useful in practical security and digital forensics scenarios.

## 4 Information-theoretic and statistical analysis: source $c(x)$ and FE output

In this section we separate two levels: (1) the perceptual binary code $C = c(X) \in \{0,1\}^n$ as a source with "noise" (robust within class, separable between classes); (2) the cryptographic artifact $(R, P) = \mathrm{Gen}(C)$ and the derived tag $T$ (via KDF/HMAC or SHA-3), for which cryptographic properties are formulated (Dodis et al., 2008; Menezes et al., 1996).

### 4.1 Properties of the perceptual source $C = c(X)$

**Bit balance and weak correlations.** For each bit $i$ we estimate the frequency $p_i = \Pr[C_i = 1]$ and pairwise correlations $\mathrm{corr}(C_i, C_j)$ (and/or mutual information). The goal is $p_i \approx \frac{1}{2}$ and small $|\mathrm{corr}(C_i, C_j)|$ for $i \neq j$. This reduces the probability of random "sticking" and increases the effective entropy of the source, and also improves the subsequent Hamming-search quality (Gong & Lazebnik, 2011; Jégou et al., 2011; Ge et al., 2013; Datar et al., 2004).

**Intra-BER and selecting radius $t$.** For admissible transforms $T \in \mathcal{T}$ we measure

$$\mathrm{BER}_{\mathrm{intra}}(x, Tx) = \tfrac{1}{n} \sum_{i=1}^{n} \mathbf{1}\{c_i(x) \neq c_i(Tx)\}.$$

We choose $t$ so that, for example, $q_{0.95}(\text{BER}_{\text{intra}}) \cdot n \leq t$—this ensures a high probability of successful reconstruction in the FE (low FRR).

**Inter distribution and rarity of perceptual collisions.** For dissimilar pairs $(x, x')$ we build the distribution $D_{\text{inter}}$ of $\text{Ham}(c(x), c(x'))/n$. Ideally, the median is close to 0.5 and the left mass (near the candidate threshold $r$) is small. For a database of size $M$, the expected number of false candidates per query is

$$\mathbb{E}[\text{FP}] \approx M \cdot p_{\leq r}, \quad p_{\leq r} = \Pr_{(x,x') \sim D_{\text{inter}}}\left[\text{Ham}(\cdot, \cdot) \leq r\right].$$

This metric controls the accuracy and load of the search stage.

**Min-entropy and leakage through public data.** We estimate a lower bound on source min-entropy as

$$H_\infty(C) = -\log_2 \max_c \Pr[C = c],$$

in practice via code/cluster frequencies or blockwise lower bounds. Publishing helper data $P$ (e.g., a linear-code syndrome of length $n - k$) reduces the available min-entropy by at most the leakage size; hence

$$\tilde{H}_\infty = H_\infty(C \mid P) \gtrsim H_\infty(C) - (n - k).$$

This quantity limits the maximum extractable secret $|R|$ and sets the security margin (Dodis et al., 2008).

**What is not claimed at the level of $c(x)$.** We do not require SAC/BIC in the strict sense and do not use the "birthday estimate" $2^{n/2}$: bits of $C$ can be dependent/biased, and the code is vulnerable to targeted adversarial manipulations. Cryptographic properties are defined only on top of the FE and the tag primitive.

### 4.2 Properties of $(R, P)$ and the cryptographic tag $T$ over the FE

**Secret extraction.** The FE guarantees that, if $\text{dist}_H(C, C') \leq t$, reconstruction $\text{Rep}(C', P) = R$ succeeds with high probability, and $R$ is statistically close to uniform on $\{0, 1\}^\ell$ given sufficient $\tilde{H}_\infty$ (closeness parameter $\varepsilon$ is chosen in the regime $2^{-\lambda}$) (Dodis et al., 2008). We then derive a key $K = \text{KDF}(R \,\|\, \text{ctx})$.

**Cryptographic tag.** The tag $T = \text{HMAC}_K(\text{meta})$ (or $T = \text{SHA3-256}(K \,\|\, \text{meta})$) inherits security (collision/second-preimage/preimage resistance) from the underlying primitive provided that $R$ is unpredictable and domain separation ctx/meta is correct (Menezes et al., 1996). Thus:

- Collision resistance of the tag reduces to the security of the chosen primitive;
- Preimage/2nd-preimage reduce to the security of the primitive and the inability to predict $R$ from $P$.

Note that tag equality means equality of the extracted secret (and thus "sufficient closeness" of codes), but not identity of the original pixels—this is an intentional FE property.

**Compatibility with search.** Candidate retrieval is performed using $c(x)$ (or its LSH signatures), while verification is done via FE→KDF→tag. Thus, both goals are preserved: fast perceptual search and cryptographically strong verification.

### 4.3 Section summary

In this section we considered two levels of the construction. At the perceptual-code level, we analyze the source $C = c(X)$ through bit balance, weak inter-bit correlations, intra-BER under admissible transforms, and the inter-distance distribution for dissimilar pairs. These

properties are intended to make the code suitable both for ANN search and as a noisy source for the fuzzy extractor.

At the FE/tag level, the goal is different: after applying the fuzzy extractor and KDF, we obtain a secret $R$ and a tag $T$ for which cryptographic guarantees are formulated in the standard way, i.e., security reduces to the underlying primitives together with the residual min-entropy budget after publishing $P$. Thus, the perceptual code itself is not treated as an independent cryptographic hash; rather, it provides a metrically robust source and index for search, while the cryptographic guarantees are assigned only to $(R, P)$ and the derived tag $T$.

In this sense, the proposed architecture connects two objectives: an avalanche-like between-class separation at the perceptual level, and standard cryptographic verification at the tag level. The current evaluation should be read as preliminary evidence that this two-layer design is workable, rather than as a complete large-scale validation.

**Applicability limitation.** Perceptual/neural hashes (our code $c(x)$) are useful for similarity search and media deduplication, but they cannot be used as a full replacement for general-purpose cryptographic hash functions (integrity, signatures). At the level of $c(x)$, targeted transforms can lead to collisions or code preservation under noticeable changes of visual content (adversarial examples) (Struppek et al., 2021; Kotenko et al., 2023) . For cryptographic purposes, (preimage/second-preimage/collision) properties in our system are provided only after applying the fuzzy extractor (obtaining $(R, P)$) and subsequent cryptographic processing (KDF, HMAC/SHA-3), in line with common cryptographic practice.

**Statistical tests (note).** Practical checks include two groups of tests. (1) For the source $C = c(X)$: bit balance, inter-bit correlations/mutual information, distributions of $\text{BER}_{\text{intra}}$ and $D_{\text{inter}}$, and a lower-bound estimate of min-entropy $H_\infty(C)$. (2) For the extracted secret $R$: randomness test batteries (e.g., NIST STS; see survey-style discussions in (Doganaksoy et al., 2010)) and checks that no degradation occurs after publishing $P$ (estimating $H_\infty(C \,|\, P)$ and the security budget). These tests do not replace a rigorous cryptanalysis, but help to detect systematic biases and leakages early.

## 5 Preliminary proof-of-concept experiments

### 5.1 Experimental setup

The current evaluation is a compact proof-of-concept intended to validate the mechanics of the proposed pipeline rather than to serve as a large-scale benchmark. We use a small demo collection of 140 images, including original images and a small number of benignly modified or stego-modified variants. The perceptual code length is $n = 512$ bits. For this collection, we evaluate: (i) intra-BER under admissible transformations, (ii) inter-image normalized Hamming distances for dissimilar pairs, and (iii) end-to-end retrieval / FE / tag-verification behavior.

### 5.2 Compact quantitative summary

### 5.3 Intra-BER and inter-distance distributions

The intra-BER histogram shows that admissible transformations usually remain within a moderate bit-error regime, which in turn determines the required error-correction capability for the fuzzy extractor. In the current experiment, the 95th percentile gives the practical estimate $t \approx 88$ bits for $n = 512$.

The inter-distance histogram shows that dissimilar-image pairs are shifted to the right relative to near-duplicate behavior, although the observed median remains below the idealized random-hash regime of 0.5. Accordingly, the current results should be interpreted as preliminary evidence of separability on a small demo dataset rather than as a final large-scale evaluation.

| Metric | Value |
|---|---|
| Dataset size | 140 |
| Code length $n$ | 512 |
| Intra-BER mean $\mu$ | 0.1466 |
| Intra-BER std $\sigma$ | 0.0260 |
| Intra-BER quantile $q_{0.95}$ | 0.1719 |
| Recommended FE radius $t$ | 88 bits |
| Inter-distance median | 0.3066 |
| Mass of inter-distances $\leq 0.3$ | 0.4255 |
| Inter-pair count | 1389 |
| Top-5 retrieved candidates verified | 3 / 5 |
| FE same-image reconstruction | success |
| FE perturbed-code reconstruction | success |

Table 1: Compact summary of the preliminary proof-of-concept evaluation.

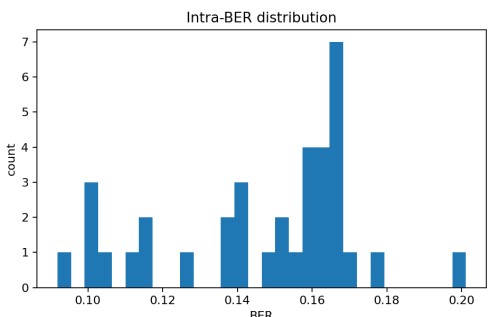 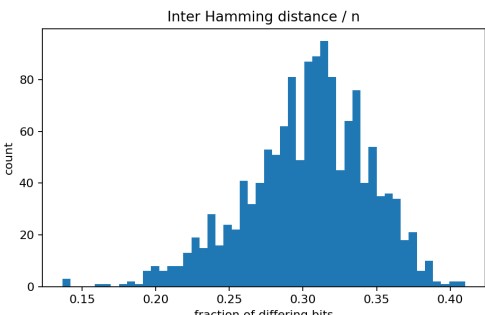

Figure 1: Left: histogram of intra-BER values for admissible transforms. Right: histogram of normalized Hamming distances for dissimilar image pairs.

## 5.4 Retrieval and verification behavior

In the retrieval experiment, the pipeline returns near candidates in Hamming space and then applies fuzzy-extractor reconstruction and tag verification. For the current top-5 retrieval example, 3 candidates pass FE/tag verification. This confirms that the end-to-end pipeline is operational as a proof of concept: perceptual retrieval is preserved, and cryptographic verification can be applied on top of retrieved candidates.

## 6 Conclusion

We presented a two-layer construction: (i) a perceptual binary code $c(x)$ obtained from a ResNet-18 embedding via projection and binarization, and (ii) a cryptographic layer based on a fuzzy extractor that produces a reproducible secret $R$ and public data $P$, from which a cryptographic tag $T$ is formed via KDF and HMAC/SHA-3. The current proof-of-concept experiments show that the pipeline is operational on a small demo dataset. For $n = 512$ bits and 140 images, we obtain an intra-BER mean of 0.1466, a 95th percentile of 0.1719 (suggesting a practical FE radius of about 88 bits), and an inter-distance median of 0.3066 for dissimilar pairs. These results indicate that near-duplicate and dissimilar pairs are separated to a useful extent for the retrieval stage, although the present evaluation remains preliminary and should not be interpreted as a large-scale benchmark.

At the level of $(R, P)$ and the tag $T$, cryptographic guarantees (preimage / second-preimage / collision resistance) reduce to the security of the employed primitives and the estimated source min-entropy after publishing $P$. At the same time, the perceptual code itself should not be regarded as a standalone cryptographic hash. Overall, the proposed architecture combines perceptual indexing with a cryptographic verification layer and is intended to

provide the practically desirable combination of two goals: fast search for similar images and a cryptographically meaningful verification tag.

Limitations remain important. The perceptual part is inherently vulnerable to targeted adversarial perturbations, and the current empirical validation is small-scale. Future work should strengthen min-entropy estimation, evaluate linkability through $P$, study larger datasets and stronger baselines, and calibrate error-correcting codes to the observed intra-BER tails.

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
