# OpenReview forum: "Implementation of a Cryptographic Hash Function Based on a Deep Neural Network."
_mathai.club/MathAI/2026/Conference — 2026 Oral_

### Official Review · Reviewer_nPyT · 2026-03-12
**Interesting approach with significant presentation issues**

**Rating:** 4
**Confidence:** 1

**Review:**

The reviewed paper presents an implementation of a cryptographic hash function based on a deep neural network, with the goal of providing fast perceptual search and cryptographically strong verification in the task of image hashing.

The strengths of the work are the theoretical foundation, the discussion of the limitations of the proposed two-layer construction, and the provided reference implementation. The English is generally correct and concise.

However, the exposition is not clear: the paper lacks an introduction, and the clearly stated goals appear on page 8 of 10. The abbreviations used (e.g., BER, SAC/BIC) are not defined; the metrics (e.g., BER, FAR) are not explained. Some steps require additional explanation or justification for readers who are not deeply versed in the field. Some parts could have been moved to the appendix to free up space for a more comprehensive exposition. The experiments (if any were conducted) are not stated in the paper itself.

The exact version of this referenced work could not be found: “T. W. Cusick and P. Stanica. Boolean functions satisfying a higher order strict avalanche criterion. In Advances in Cryptology – EUROCRYPT ’93, volume 765 of Lecture Notes in Computer Science, pp. 77–87. Springer, 1994.” There is, however, a version without P. Stanica as a co-author and with different pages (pp. 102–117): “Cusick, T.W. (1994) Boolean Functions Satisfying a Higher Order Strict Avalanche Criterion. In: Helleseth, T., Ed., Advances in Cryptology—EUROCRYPT’93. EUROCRYPT 1993. Lecture Notes in Computer Science, Vol. 765, Springer, Berlin.”

Additionally, there are several formatting issues:
- Headers are not in small caps.
- Several incorrect citations: citations in parentheses when the publication is included in the sentence (e.g., at p. 4: “see (Dodis et al., 2008)”), one missing or mistyped citation (p. 2: “(?Webster & Tavares, 1986; Cusick & Stanica, 1994; Menezes et al., 1996)”). The URL of “Handbook of Applied Cryptography” by Menezes et al. links to a chapter of the book and not the book itself.
- Many instances of “/” should be replaced with conjunctions for better readability (e.g., “approximate independence/balance of bits”).

While promising, the proposed work is not yet ready for publication. It is recommended to edit the article again to ensure compliance with the submission guidelines. The article would potentially benefit from the inclusion of the proposed implementation as a pseudocode.

---

> ### Author Rebuttal · Authors · 2026-03-12
>
> Thank you to the reviewer for the careful reading of the paper and for the constructive comments.
>
> We agree with the main concerns regarding the presentation. In the current version, the problem setup is not introduced clearly enough: the introduction and the explicit statement of goals should appear much earlier and be made more accessible to the reader. We also agree that the abbreviations and metrics used in the paper, in particular BER, FAR, and SAC/BIC, should be explicitly defined upon first use. These changes will be made in the revised version of the paper.
>
> Regarding the experiments, they were in fact conducted, but they were not presented properly in the submitted manuscript. The implementation and experimental part currently exist in a separate repository / notebook and were not included as part of the anonymized submission. We agree that this gives the impression that empirical validation is missing, and in the revised version we will add a compact quantitative summary of the experimental setup and results.
>
> We also thank the reviewer for pointing out the formatting and bibliography issues. All noted problems — abbreviation definitions, introduction structure, citation style, header formatting, and bibliographic corrections — will be addressed.
>
> Overall, we agree that the current submission suffers primarily from presentation issues, and we appreciate the reviewer for highlighting them. We will incorporate these improvements in the revised version.

---

### Decision · Program_Chairs · 2026-03-14

**Decision:**

Accept (Oral)

**Comment:**

Dear Author(s),

On behalf of the Program Committee of the International Conference on Mathematics of Artificial Intelligence (MathAI 2026), we are pleased to inform you that your paper has been accepted for an oral presentation at MathAI 2026.

Your paper was evaluated through a rigorous two-stage review process involving both automated screening and expert review by members of the Program Committee. The reviewers recognized the quality and contribution of your work.

Presentation details:

- Format: Oral presentation (15–20 minutes + 5 minutes Q&A)
- Mode: You may present either in person (offline) at the conference venue in Sirius, Russia, or remotely via Zoom. Please indicate your preferred mode when confirming your participation.
- Conference dates: Marh 30 - April 3, 2026
- Website: https://mathai.club

Next steps:

1. Please confirm your participation and presentation mode by replying to this email mathai.club@yandex.ru no later than March 15, 2026 18:00 Moscow time.
2. If you plan to attend in person, the organizing committee will provide accommodation details separately.
3. Please prepare your final camera-ready manuscript according to the formatting guidelines available at https://mathai.club and upload it to OpenReview by March 15, 2026 18:00 Moscow time.

Should you have any questions regarding the program, logistics, or your presentation slot, please do not hesitate to contact us.

We look forward to your contribution to MathAI 2026.

With kind regards,

MathAI 2026 Program Committee
International Conference on Mathematics of Artificial Intelligence
https://mathai.club
OpenReview: https://openreview.net/group?id=mathai.club/MathAI/2026/Conference
Telegram: https://t.me/MathAI_club
Email: mathai.club@yandex.ru